# The Complete Chloroplast Genome Sequencing and Comparative Analysis of Reed Canary Grass (*Phalaris arundinacea*) and Hardinggrass (*P. aquatica*)

**DOI:** 10.3390/plants9060748

**Published:** 2020-06-14

**Authors:** Yi Xiong, Yanli Xiong, Shangang Jia, Xiao Ma

**Affiliations:** 1College of Animal Science and Technology, Sichuan Agricultural University, Chengdu 611130, China; xiongyi95@126.com (Y.X.); yanlimaster@126.com (Y.X.); 2College of Grassland Science and Technology, China Agricultural University, Beijing 100193, China; 3Key Laboratory of Pratacultural Science, Beijing Municipality, Yuanmingyuan West Road, Haidian District, Beijing 100193, China

**Keywords:** *Phalaris arundinacea* L., *Phalaris aquatica* L., chloroplast genome, high-throughput sequencing, RNA-seq, ploidy

## Abstract

There are 22 species in the *Phalaris* genera that distribute almost all over the temperate regions of the world. Among them, reed canary grass (*Phalaris arundinacea*, tetraploid and hexaploid) and hardinggrass (*P. aquatica*, tetraploid) have been long cultivated as forage grass and have received attention as bio-energy materials in recent years. We aimed to facilitate inter-species/ploidies comparisons, and to illuminate the degree of sequence variation within existing gene pools, chloroplast (cp) genomes of three *Phalaris* cytotypes (*P*. aquatica/4x, *P*. *arundinacea/*4x *and P.*
*arundinacea*/6x) were sequenced and assembled. The result indicated that certain sequence variations existed between the cp genomes of *P. arundinacea* and *P. aquatica.* Several hotspot regions (*atpI*~*atpH*, *trnT-UGU*~*ndhJ*, *rbcL*~*psaI*, and *ndhF*~*rpl32*) were found, and variable genes (*infA*, *psaI*, *psbK*, etc.) were detected. SNPs (single nucleotide polymorphisms) and/or indels (insertions and deletions) were confirmed by the high Ka/Ks and Pi value. Furthermore, distribution and presence of cp simple sequence repeats (cpSSRs) were identified in the three *Phalaris* cp genomes, although little difference was found between hexaploid and tetraploid *P. arundinacea*, and no rearrangement was detected among the three *Phalaris* cp genomes. The evolutionary relationship and divergent time among these species were discussed. The RNA-seq revealed several differentially expressed genes (DEGs), among which *psaA*, *psaB*, and *psbB* related to leaf color were further verified by leaf color differences.

## 1. Introduction

To fully understand the evolutionary paths of *Phalaris* species, it is necessary to know the information coded in their nuclear genome and the accompanying organelle genomic (mitochondria and/or chloroplast) sequence [1]. Given that the nuclear genome assembly is costly and difficult, the genome sequences of mitochondria and chloroplast are much smaller and thus appropriate for genetic information mining. Chloroplast (cp) genome sequences have been proven useful in genetic studies like differential gene detection, polymorphic probe development, and phylogenetic analysis [1]. Mutation or structural variation in cp genome is of great significance in studying plant evolution, classification, genetic diversity, mechanism of plant photosynthesis, plant energy metabolism, plant antioxidants, and secondary metabolism [2]. Especially, the cp genes in plants are specifically inherited from the maternal line and not disturbed by gene recombination; hence, the evolutionary route is relatively independent from the nuclear genome. This makes it good material for the study of plant genealogy geography, systematics, gene and intron losses, and population history, as well as excellent substrates for genetic transformation [1,2,3]. A typical plant cp genome size ranges from 120 kb to 217 kb. It usually has a quadripartite structure consisting of two copies of an inverted repeat (IR) region, a large single-copy (LSC) region, and a small single-copy (SSC) region [1,4]. Recent studies showed that the non-coding intergenic spacer regions of the cp genome have considerable diversity and may function in regulation of gene expression [5,6].

*Phalaris* L. (Poaceae) wildly distributes in temperate regions of both hemispheres and in the mountains of tropical Africa and South America. At least 22 *Phalaris* species have been identified [7,8,9,10]. Among them, *Phalaris aquatica* and *P. arundinacea* both originated in the Mediterranean region and are closely allied with each other in the lineage with x = 7 basic chromosomes. The *P. arundinacea* (6x) was reported genetically closer to *P. aquatica* (4x) than to *P. arundinacea* (4x) because of a higher hybridization rate between *P. arundinacea* (6x) and *P. aquatica* (4x) [10]. Hence, the hypothesis was once made that *P. arundinacea* (6x) was the outcome of the hybridization of *P. aquatica* (4x) and *P. arundinacea* (4x). However, more genetic and fossil evidence is needed to support this hypothesis [11].

*Phalaris* species are not only very important for grazing, but also receive close attention as a biomass and bio-energy crop which showed great yield potential in consideration of both dry matter and crude protein via seed dispersal or extensive rhizome systems [9,12]. Moreover, the *Phalaris* species possess adaptation to cold, drought, and flooding, as well as all kinds of soils; therefore, they are mainly planted on marginal land that is unsuitable for other agricultural uses [7,10]. Additionally, because *Phalaris* have evident variation in morphological traits, plus the vital role of polyploidy in their evolution (e.g., tetraploid and hexaploid *P. arundinacea*), their potential value in terms of speciation, chromosomal evolution, and biogeography in Poaceae were manifested [8]. The problem is, however, that most available accessions do not exhibit good palatability due to the existence of poisonous alkaloids, poor digestibility in comparison with other forage species, and low seed yield because of high seed shattering [13]. According to previous studies, alkaloid concentration of *P. arundinacea* is related to the moisture stress and nitrogen application, as well as clipping and light intensity, which react as a response to the external environmental changes and work as stimulating or regulating factors in plant growth, metabolism, and reproduction [13,14,15]. Moreover, alkaloid in *P. arundinacea* was reported to be accumulated mainly in the leaf blades with a percentage of more than 59% of the concentration [13]. A previous study on lupins also indicated that the lysine-derived quinolizidine alkaloids appeared to occur within the mesophyll chloroplasts of green leaves [16]. Hence, we believe that more or less hereditary factor controlling alkaloid synthesis lies in the cp genome of *Phalaris* or resulted from the interaction of nuclear and plastids genes. In this case, cp genome modification of *Phalaris* is feasible because engineering the cp genome for biotechnology applications was reported to succeed in many other species [4].

Identification of genomic variation provides an opportunity for the uncovering of correlation between genotype and phenotypes [17]. Given the fully transcribed character of the cp genome [18], transcriptome sequencing (RNA-seq), along with cp genome sequencing, can be considered as a powerful tool to gain functional genomic data and to identify the function of candidate cp genes in specific biological process [17,18,19]. That will supply valuable information for further gene function research.

In this study, cp genomes of two *Phalaris* species were sequenced and compared. We revealed sequence synteny and rearrangements, gene loss/pseudogenization, and IR expansions and contractions among these species. Divergence times of the sequenced *Phalaris* species from other complete sequenced Pooideae grass were estimated. Our analyses of the three *Phalaris* cp genomes (*P. arundinacea* (tetraploid and hexaploid) and *P. aquatica* (tetraploid)) provide detailed genetic information distinguishing different *Phalaris* ploidies and facilitate breeding programs and species identification. In addition, RNA-seq was carried out to detect the differentially expressed genes (DEGs) and to explore their corresponding functions.

## 2. Results

### 2.1. Genomic Features of Three Phalaris Chloroplast Genomes

The assembled cp genomes of *P*. a*rundinacea* (4x), *P*. a*rundinacea* (6x), and *P*. *aquatica* (4x) were 135,898 base pairs (bp), 135,910 bp, and 136,249 bp, respectively (Table 1). All of them had a typical quadripartite structure. The pair of inverted repeats (IRs, including IRA and IRB) regions of the three cp genomes were 21,653 bp, 21,653 bp, and 21,703 bp, separated by an LSC region and an SSC region (Table 1 and Figure 1). The guanine and cytosine (GC)contents of the cp genomes were 38.52%, 38.52%, and 38.45%, and the percentages of GC nucleotides were unevenly distributed throughout the cp genomes of those with the highest GC content in IRs.

The total numbers of unique cp genes were 104, 104, and 105, respectively, in the three *Phalaris* cp genomes. *P. arundinacea* (4x) and *P. arundinacea* (6x) both had 28 tRNAs, 4 rRNAs, and 72 mRNA genes (Table 1), while one additional gene, *ycf4*, was found in *P. aquatica* (Table 2). The three cp genomes all contain 19 genes duplicated in the IR region, including all four kinds of rRNAs (16S, 23S, 5S, 4.5S), seven tRNAs, six translation-/transcription-/photosynthesis-related genes, *ycf1*, and *ycf2*. The 72 common mRNA genes in the three cp genomes consisted of 24 unique transcription- and translation-related genes, 43 (*P. arundinacea*) and 44 (*P. aquatica*) unique photosynthesis-related genes, and five other genes: *matK*, *clpP*, *cemA*, *ycf1*, and *ycf2* (Table 3).

In all the three *Phalaris* cp genomes, 12 genes had one single intron (Table 2), which was highly conserved in the two *P. arundinacea* ploidies, and small differences were observed for intron size between *P. aquatica* and *P. arundinacea*. The intron-containing genes could be categorized into three types corresponding to electron transfer, protein synthesis, and ATP synthesis (Table 2 and Table 3).

### 2.2. Variation among Three Chloroplast Genomes

Overall genic variation among the three *Phalaris* cp genomes was revealed by mVISTA [20] and Mauve [21]. We found more conservation in the coding regions than that in non-coding regions, and higher divergence in LSC regions than in SSC and IR regions (Figure 2). The cp genomes of the two different ploidy levels of *P. arundinacea* species showed high conservation, while *P. aquatica* varied from *P. arundinacea*, especially in the non-coding sequence of LSC regions. Hotspot regions highly enriched with variations were identified in the whole genome, which included *atpI*~*atpH*, *trnT-UGU*~*ndhJ*, *rbcL*~*psaI*, *ndhF*~*rpl32*, etc. (Figure 2). However, no rearrangement or inversion events were found among the three *Phalaris* cp genomes as depicted in the locally collinear blocks (LCBs) (Appendix A).

The indels (insertions and deletions) and SNPs (single nucleotide polymorphisms, including Tv (transversion) and Tn (transition)) were identified among the three *Phalaris* cp genomes using mafft software [22] (Appendix A). In total, 98 and 95 indels were predicted in comparisons of *P. aquatica* (4x) vs. *P. arundinacea* (6x), and *P. aquatica* (4x) vs. *P. arundinacea* (4x), among which 6 indels were observed in the coding sequences. There were 14 indels between *P. arundinacea* (6x) and *P. arundinacea* (4x) existing in the noncoding sequence (Appendix A). Similarly, there was approximately the same number of Tv and Tn in *P. aquatica* vs. *P. arundinacea* (6x) (Tv = 399, Tn = 85) and *P. aquatica* vs. *P. arundinacea* (4x) (Tv = 397, Tn = 77). It is worth noting that both Tv and Tn were dominantly located in the intergenic region, and more Tvs were found than Tns in both the genic and intergenic regions (Figure 3A,C). Additionally, we identified many more SNPs than indels in *P. aquatica* vs. *P. arundinacea*. However, no SNPs were found in the genic region in a comparison of the two *P. arundinacea* ploidies.

SNPs and indels were also counted in the quadripartite structure (LSC, SSC, and IRs, Figure 3). Obviously, the variations occurred mainly in the LSC region when *P. aquatica* and *P. arundinacea* were compared (Figure 3B,D). However, in the two *P. arundinacea* ploidies (4x vs. 6x), there were no SNPs/indels in the genic region (Figure 3E) and no indels in the SSC and IR regions (Figure 3F). The genes of *matK*, *rpoB*, and *rpoC2* all contain more than ten variations (indels and/or SNPs), indicating high divergence.

We used the 72 common protein-coding genes in the three cp genomes to calculate non-synonymous (Ka) and synonymous (Ks) rates and the Ka/Ks ratio (Appendix A). The result showed that the values of Ka/Ks for most of those genes were < 1 or failed to be calculated because of Ka or Ks = 0 between *P*. *aquatica* and *P*. *arundinacea*, which indicated their conservation without any non-synonymous or synonymous nucleotide substitution. However, the gene *rbcL* had a Ka/Ks > 1 between *P*. *aquatica* and *P*. *arundinacea*, suggesting a possible positive selection. There were no non-synonymous substitutions between *P*. *arundinacea* (6x) and *P. arundinacea* (4x). In addition, the identified non-synonymous substitutions between *P*. *aquatica* and *P*. *arundinacea* were involved in 14 (6x) and 15 (4x) genes (Appendix A).

### 2.3. IR Scope Characteristics

The shrinkage and expansion of IR regions in the three *Phalaris* cp genomes were compared with each other. As shown in Figure 4, there was a small difference at the junction positions among the three acquired cp genome sequences. The genes *ndhF*-*ndhH* and *rpl22*-*rps19*-*psbA* were located at the junctions of IR and single copy regions. The *ndhH* gene spanned the SSC/IRA region with 181 bp located at the IRB region. Spacers ranged from 36 (*P. arundinacea*) to 38 bp (*P. aquatica*), separating the *rps19* gene from the LSC/IRA border and from the LSC/IRB border. In addition, a 94 bp spacer separated the *ndhF* gene from the SSC/IRB border in all three *Phalaris* species.

### 2.4. Nucleotide Diversity

Genetic distance among *Phalaris* cp genomes was evaluated by the calculation of nucleotide diversity (Pi) of 107 common genes unique in every quadripartite region using DnaSP v5.0 software [23]. The mean Pi value of three *Phalaris* cp genomes was 0.01628, showing a high level of intergeneric differentiation. The SSC region exhibited the highest nucleotide diversity in view of its average Pi value (PiSSC = 0.04606), while the IR regions had the lowest level of Pi (PiIR = 0.00012), suggesting a much higher conservation of the IR region (Appendix A). The comparison of three *Phalaris* cp genomes identified seven genes with Pi > 0.006. Two genes, *psaC* and *rpl32*, were located in the SSC regions, and the other five (*infA*, *psaI*, *psbK*, *rpl16*, and *rpoB*) were located in the LSC region.

### 2.5. Repetitive Sequences 

The analysis of repetitive sequences identified 186, 181, and 180 perfect simple sequence repeats (SSRs) in total in the cp genomes of *P*. *aquatica*, *P*. *arundinacea* (6x), and *P*. *arundinacea* (4x), respectively. Most SSRs were found to be at the LSC region within its intergenic areas, while there were no SSRs found in intron and intergenic portion within SSC region. We found that the percentage of SSRs was higher at the LSC or SSC region, but lower at the IR regions in *P*. *aquatica* than those of the others (Figure 5A). Among these SSRs, all kinds of repeats, including mononucleotides, di-, tri-, tetra-, and pentanucleotides, were found to exist in either *P*. *aquatica* or *P*. *arundinacea.* The exception of hexanucleotides was only found in the two *P*. *arundinacea* ploidies (Figure 5B).

In total, 173 (*P*. *aquatica*), 185 (*P*. *arundinacea*/6x), and 179 (*P*. *arundinacea*/4x) pairs of repeats (15 bp or longer) were detected using the program REPuter [24]. There were nearly equal numbers of forward and palindromic repeats in all these *Phalaris* cp genomes (Figure 5C). Additionally, the result showed that 15–22 bp repeats, as well as 24 bp, 26 bp, 29 bp, and 30 bp repeats, occurred in all three chloroplast genomes, while 23, 28, 33, 34, 73, and 212 bp repeats were only detected in the *P*. *aquatica* cp genome, and 286 bp long repeats were only found in *P*. *arundinacea* (6x and 4x) cp genomes (Figure 5D).

### 2.6. Codon Usage

The codon usage frequency and relative synonymous codon usage (RSCU) were thought to be a combination of natural selection, mutation, and genetic drift. Here, the RSCU values were calculated based on 72 protein-coding genes of the three *Phalaris* cp genomes. The most abundant amino acid was leucine, whose numbers of codons were 2097 (10.866%), 2098 (10.872%), and 2125 (10.862%) in *P*. *arundinacea* (4x), *P*. *arundinacea* (6x), and *P*. *aquatica*, respectively. The least abundant amino acid was cysteine, which possessed 211 (1.079%), 208 (1.078%), and 208 (1.078%) codons in each genome, respectively (Appendix A). Codon usage was observed biased towards A/U-ending codons (RSCU > 1) in the three cp genomes, which was consistent with the study of angiosperm cp genome [25]. In addition, we found that the only codon that had no bias in the *Phalaris* species (RSCU = 1) was UGG, which encoded tryptophan (Appendix A).

### 2.7. Phylogenetic Divergence Time

Evolutionary divergence of the two *Phalaris* species along with other 12 Poaceae species were estimated based on their cp genomes. Obviously, the phylogeny of these species was in consistence with the classical botanical classification (Figure 6). As expected, *P. aquatica*, *P. arundinacea* (6x), and *P*. *arundinacea* (4x) clustered together in the *Phalaris* clade. The divergence time between *P*. *aquatica* and *P. arundinacea* was about 3.3055 million years (Mya), and *P*. *arundinacea* (6x) and *P*. *arundinacea* (4x) were separated from each other about 0.1446 Mya ago.

### 2.8. Diversity and Expression of Chloroplast Genes

We employed RNA-seq to look into the cp gene expressions. The result showed the transcript abundances of cp genes ranged from 3.0 to 2602,005.9 FPKMs (fragments per kbp of gene per million mapped reads); especially the *psbM* gene showed a much higher level of expression in all cp genomes, followed by *psaC*, *psbD*, and *psbA* (Appendix A). The inter-genome differentially expressed genes (DEGs) were identified based on the fold change (FC ≥ 2), and most of the up-regulated DEGs in *P. arundinacea* (6x) vs. *P. arundinacea* (4x) and *P. arundinacea* (6x) vs. *P. aquatica* were tRNA-/rRNA- genes, especially *trnfM-CAU* with a high FC value. In addition, three genes of *psaA*, *psaB*, and *psbB*, which were annotated with a gene ontology (GO) molecular function of chlorophyll binding and photosynthesis, were found up-regulated in *P. aquatica*, compared to *P. arundinacea* (6x) and *P. arundinacea* (4x) (Appendix A). This result was further verified by the darker green leaf color of *P. aquatica* (Figure 7B). The relative chlorophyll content also indicated a significant difference between *P. aquatica* and *P. arundinacea* (Figure 7A).

Meanwhile, abundant variations were also detected within each *Phalaris* cp genome applying RNA-seq dataset, among which RNA editing (RE) was dominant. With a cut-off value of at least 5% edited reads, 407, 76, and 83 RE sites were identified in total in the cp genomes of *P. aquatica*, *P. arundinacea* (4x), and *P. arundinacea* (6x), respectively, among which 214 (52.58%), 54 (71.05%), and 56 (67.47%) were located in the genic region (Appendix A), and the nonsynonymous mutation rates in genic regions were 80.84%, 22.22% and 26.79%, respectively.

## 3. Discussion

### 3.1. Chloroplast Genome Characteristics of Three Phalaris Cultivars

The cp genomes of *Phalaris* species with different ploidies were assembled and compared to each other in this study. Interestingly, more differences were observed between *P*. *aquatica* and *P*. *arundinacea* species but not between different ploidy of *P*. *arundinacea*. The sizes of the whole cp genome, LSC, SSC, and IRs of *P*. *aquatica* were larger than those of the two *P*. *arundinacea* cp genomes, while the GC content in these regions were a little bit lower in *P*. *aquatica*. Although cp genomes among Poaceae are highly conserved, the existing differences would provide insight into unique variations among species or subspecies [5,6]. Furthermore, there was higher amino acid abundance of Phe, Tyr, Trp, and Lys in *P*. *aquatica* than that in both tetraploid and hexaploid *P*. *arundinacea* (Appendix A), which might be related to alkaloid synthesis [26]. According to Vincenzo et al. [26], the quinolizidine skeleton is constructed in chloroplast, although the further modifications of Lys-obtained quinolizidine alkaloids should be completed after intracellular transport to the cytosol and mitochondria. Therefore, the genes of chloroplast might participate in this biological process of alkaloid synthesis. The *ycf4* gene, as a photosystem I assembly protein, was unique in *P*. *aquatica* while absent in *P*. *arundinacea*. It suggested a more stable accumulation of photosystem I complex within the thylakoid membranes [27], and the *ycf4* gene might play an important role in biosynthesis process of different alkaloids.

### 3.2. Sequences Variation and Gene Mutation

Several protein-coding genes (*infA*, *psaI*, *psbK*, *rpl16*, *rpoB*, *psaC*, and *rpl23*) showed high Pi value would have the potential to be good markers for DNA barcoding and phylogenetic analyses, and to expand the cytoplasmic gene pool related to yield enhancement, seed retention, alkaloid reduction, and so on. It is remarkable that the *infA* gene was found in all the three *Phalaris* cp genomes, as it is commonly lost in angiosperm groups due to its hydrophilicity [28]. These hotspot regions with mutation events would provide crucial genetic information for the further research in the evolutionary history of *Phalaris* or Poaceae species [1]. IR-size variation and gene loss are of the main reasons for variation of the cp genome size. As inverted repeats extended into neighboring single copy regions, differences occurred near the boundary of the IR/SC. These are possible reasons for gene fragmentation and incorporation of large chunks of single copy regions within the IR [1]. However, no rearrangement was found among the three *Phalaris* cp genomes, while the extension of IRs/SC boundaries neighboring *ndhH* and *rps19* in the IR regions was observed. Although expansions and contractions of the IRs/SC junctions in angiosperm cp genomes are well investigated, the ones in the Pooideae are unique and still useful [29,30] for the evolutionary studies.

The Ka/Ks value is an important indication of natural selection and molecular sequence evolution, which is a result of both frameshift and missense mutations [31]. Comparison of SNPs and indels among the three cp genomes indicated that they occurred mainly in the intergenic or LSC regions. SNPs or indels between *P*. *aquatica* and *P*. *arundinacea* (6x) or *P*. *arundinacea* (4x) were roughly similar to each other, while those between two *P*. *arundinacea* cp genomes showed a different pattern. No SNPs or indels were found in the genic or IRs regions between the two *P*. *arundinacea* cp genomes, which indicated that IR regions were the most conservative quadripartite structure in the cp genome, and the detailed reason is still unknown [1]. 

### 3.3. CpSSRs and RSCU

One of the most valuable tools in population genetics is cpSSRs marker, which has unique characters such as, non-recombination, haploidy, and so on [32]. Several cpSSRs identified in this study could be used for further genetic polymorphism detection of *Phalaris* species. The RSCU is thought to be influenced by a combination of natural selection, mutation, and genetic drift [33]. In this study, RSCU value was calculated using common coding genes of three *Phalaris* cp genomes, and our result indicated that the most and least abundant amino acids were leucine and cysteine, respectively, which is consistent with the previous study in the angiosperm cp genomes [34]. Besides, we found that almost all the A/U-ending codons hold a bigger usage frequency (RSCU > 1), and the C/G-ending codons were on the contrary, suggesting a significant avoidance of C/G pairs in highly conserved regions [25].

### 3.4. Phylogeny Analysis and Divergence Time

We conducted a phylogenetic analysis based on the three whole *Phalaris* cp genomes and other Pooideae species, and the phylogenetic clades in Bayesian tree were in consistence with the classical botanical taxonomy. The divergence of *P*. *aquatica* and *P*. *arundinacea* was estimated to be approximately 3.3055 million years (Mya) ago with the arrival of quaternary glacial period, which may partly explain why the tetraploid *P*. *arundinacea* is more cold-resistant [10,35]. Furthermore, our result showed a closer maternal relationship of hexaploidy *P*. *arundinacea* to tetraploidy *P*. *arundinacea* than to tetraploidy *P*. *aquatica*, though a closer kinship on the production traits between *P*. *arundinacea* (6x) and *P*. *aquatica* was reported [10], which were probably caused by the similarities of nuclear genomes rather than maternal cp genomes.

### 3.5. Gene Expression in Chloroplast Genomes

RE sites were considered as mismatches between RNA-seq reads and the cp genome sequence, that would account for the majority of within-genome polymorphism and indicated multi-haplotypes in each of the three sequenced *Phalaris* cp genomes [36]. The RE phenomenon was reported to enlarge the genetic information and make organisms better adapted to their environment [36]. Given the same germination and growth condition until sampling, the within-genome variation of the two *Phalaris* species may be inherited from their ancestors or caused by the nonconservative replication of enzymes. In view of the cp genes expression pattern revealed by RNA-seq, the down-regulated DEGs were almost related to replication and transcription function in *P. aquatica* compared to *P. arundinacea* (4x) and *P. arundinacea* (6x), while the up-regulated DEGs were mostly enriched in photosynthesis, energy metabolism, and chlorophyll synthesis. In addition, the RE pattern of *P. arundinacea* (6x) and *P. arundinacea* (4x) were more alike, as the cp DNA of *P. arundinacea* (6x) may be inherited from *P. arundinacea* (4x) in maternal inheritance way [32]. The leaf color of three *Phalaris* cultivars was observed different, and the related genes *psaA*, *psaB,* and *psbB* are worth to be studied with regard to their significantly increased expression in *P. aquatica*.

## 4. Materials and Methods

### 4.1. DNA Extraction and Genome Sequencing

Three *Phalaris* cultivars, *Phalaris arundinacea* L. (including tetraploid and hexaploid) and *Phalaris aquatica* L. (tetraploid), were all NPGS (National Plant Germplasm System of the United States) collections (Table 4). Totally 100 mg fresh leaves from seed germination were applied for total genomic DNA extraction using the Plant DNA Extraction Kit (Tiangen, Beijing, China). The DNA quality were verified by 1% agarose gel before libraries construction and library quality testing. Finally, total genomic DNA sequencing was conducted on the Illumina Novaseq PE150 platform.

### 4.2. Chloroplast Genome Assembly and Annotation

The three studied *Phalaris* cp genomes were assembled using SPAdes v3.10.1 (Saint Petersburg state University, Saint, Russia) [37] and Gapfiller v2.1.1 (BaseClear BV, Einsteinweg, Leiden, The Netherlands) [38] software, based on the reference of *Phalaris arundinacea* (NC027481.1, unknown ploidy) cp sequence [39] from NCBI. The SEED sequence of cp genome was firstly obtained by assembling cp DNA sequence with SPAdes software. The pseudo contigs were identified through kmer iterative extend seed. Then the obtained contig sequences were connected to obtain scaffolds via SSPACE v2.0 (BaseClear BV, Einsteinweg, Leiden, The Netherlands) [40]. The Gapfiller v2.1.1 procedure was used to make up the gaps and then the GC content was calculated. On the other hand, Blast v2.2.25 (U.S. National Library of Medicine 8600 Rockville Pike, Bethesda MD, 20894 USA) [41] pipeline was used to compare against the chloroplast genome coding sequences (cds) in NCBI, and the final gene annotation results of cp genome were obtained after manual correction. Hmmer v3.1b2 (HHMI/Harvard University, Boston, USA; The European Bioinformatics Institute, Cambridge, UK) [42] software was used to compare the rRNA sequence of the above chloroplast genomes. Aragorn v1.2.38 (Murdoch University, Western Australia, Australia; Lund University, Lund, Sweden) [43] software was used to predict the tRNA sequence of chloroplast genomes. A circular cp genome map of *Phalaris* was drawn with the Organellar Genome DRAW 1.3.1 (Max-Planck-Institut fur Molekulare Pflanzenphysiologie, Am Muhlenberg 1, D-14476 Potsdam-Golm, Germany) [44].

### 4.3. Multiple Chloroplast Genome Alignments

The mVISTA (National energy research scientific computing center, San Francisco, USA) [20] program with LAGAN mode (Department of Computer Science, Stanford University, California, USA) was applied to do multiple alignments between the three studied *Phalaris* cp genomes, using the cp genome of NPGS *Phalaris arundinacea* (NC027481.1) as a reference. Synteny and rearrangement analysis of above-mentioned species were performed based on Mauve [21] pipeline. Variation in size of IR regions is crucial in evolution, the boundaries in the junction of IR and SC regions of these three cp genomes were compared using the IRscope (University of Helsinki, Helsinki, Finland) [45] online software. Nucleotide diversity (Pi) can reveal the variation of nucleic acid sequences in different species, and the sites with high variability indicated potential molecular markers for population genetics. Here, Mafft 7.037 (Pennsylvania State University, Pennsylvania, USA; Kyoto University, Kyoto, Japan) [22] software was used to conduct global comparison of cds sequences of the common genes in different *Phalaris* cp genomes, and DNAsp 5.0 (Universitat de Barcelona, Barcelona, Spain) [23] was used to calculate the Pi value of each gene.

### 4.4. Identification of Repetitive Sequences

MISA v1.0 (Leibniz Institute of Plant Genetics and Crop Plant Research (IPK) Gatersleben, Corrensstr. 3, 06466 Seeland, Germany) [46] software was used to identify chloroplast Simple Sequence Repeats (cpSSRs) following the parameter that ‘1-8’ mean single-base repeat 8 or more times, ‘2-5’ means double bases repeat 5 or more times, and ‘3-3’, ‘4-3’, ‘5-3’, ‘6-3’ may be deduced by analogy. In addition, interspersed repetitive sequences, that distributed in a disperse way throughout the genome, including direct (forward), inverted (palindromic), complement and reverse repeats, were searched by REPuter 3.0 (University of Bielefeld, Bielefeld, Germany; Max Planck institute for molecular genetics, Berlin, Germany) [24] software with a minimum repeat size of 15 bp and sequence identity greater than 90%.

### 4.5. Relative Synonymous Codon Usage Analysis

Because of the degeneracy of codons, each amino acid was coded by one to six codons. The utilization rate of genomic codon varies greatly among different species and organisms. This unequal use of synonymous codons is known as Relative Synonymous Codon Usage (RSCU), which was thought to be influenced by natural selection, mutation, and genetic drift. The RSCU analysis was performed using CodonW 1.4.4 (University of Nottingham, Nottingham, UK) software [16].

### 4.6. Analysis of Non-Synonymous/Synonymous Substitution

When base variation results in changes in amino acids, it is non-synonymous substitution (Ka), or else, synonymous substitution (Ks). Non-synonymous mutations are generally affected by natural selection. The ratio of non-synonymous mutation rate to synonymous mutation rate (Ka/Ks) indicates the selection effect: Ka/Ks ratio > 1, indicated positive selection effect; Ka/Ks ratio < 1, indicated purification selection effect; and Ka/Ks ratio = 1, mean neutral evolution. Here, mafft 7.037 software [22] was used to perform gene (protein-coding exon) sequence alignment of three *Phalaris* cp genomes, and the Ka/Ks rates of these common coding genes were calculated using the KaKs Calculator v2.0 (CAS Key Laboratory of Genome Sciences and Information, Beijing Institute of Genomics, Chinese Academy of Sciences, Beijing 100029, PR China; Plant Stress Genomics Research Center, Division of Chemical and Life Sciences & Engineering, King Abdullah University of Science and Technology, Thuwal 23955-6900, Kingdom of Saudi Arabia) [47].

### 4.7. Phylogenetic Analysis

Twelve published Poaceae genomes in NCBI (Appendix A) and three *Phalaris* cp genomes sequenced in this study were collected to conduct the phylogenetic analysis, with *Bambusa multiplex* (NC024668.1) and *Panicum virgatum* (HQ731441.1) as outgroups [8]. BEAUti (Department of Computer Science, University of Auckland, Auckland, New Zealand) was applied to construct a Priors tree following the strict clock and Yule model under GTR + G + I substitution assumption [48]. The MCMC setting was 10,000,000 of Chain length, 1000 of Tracelog, 1000 of screenlog and 1000 of treelog. Tracer v 1.5 (Institute of Evolutionary Biology, University of Edinburgh, Edinburgh, UK) [49] was performed to test the value of effective sample size (>200). Finally, the tree was visualized in Figtree v1.4.3 (Institute of Evolutionary Biology, The University of Edinburgh, Edinburgh Scotland) [50].

### 4.8. Chloroplast RNA-Seq and Chlorophyll Measurement

For transcriptome analysis, fresh young plant leaves of each accessions were cut and used to construct cDNA libraries, respectively. Total RNA was extracted using Trizol Reagent and treated with RNase-free DNase I (TIANGEN BIOTECH CO., LTD, Beijing, China), and then a NanoDrop ND1000 spectrophotometer (Thermo Scientific, Wilmington, DE, USA) along with 1% agarose gel electrophoresis were used for quality examination of the total RNA. The sequencing of RNA samples was performed in Illumina Novaseq PE150 platform by Biomarker Technologies (Beijing, China). FPKM values for cp genes of each accession were used for transcript abundance calculation by mapped the RNA-seq reads to their own cp genome.

The relative chlorophyll (chl) content of the leaves was recorded with a chl content colorimeter CL-01 (Hansatech Instruments Ltd., Norfolk, UK). The instrument determines the relative chl content on the basis of dual wavelengths of the spectral absorbance at 620 and 940 nm [51]. Three clones of each *Phalaris* accession were chosen in vegetative growth stage which was the same to that of RNA-seq sampling. In each plant, ten leaves from the apex of different tillers were randomly selected to measure at base of leaf blade. Significant differences for the relative chlorophyll average values were determined based on the least significant difference (LSD) test at *p* ≤ 0.05. 

## 5. Conclusions

The cp genomes of two *Phalaris* species *P. aquatica* and *P. arundinacea* (including tetraploid and hexaploid) were firstly sequenced and annotated in this study, which was beneficial to the evolutionary study of *Phalaris* genus. Though gene content were highly conserved and no arrangement among those three *Phalaris* cp genomes, some hotspot regions and variable genes were identified. Furthermore, the divergence time and evolutionary relationship among these three *Phalaris* cp genomes along with other Gramineae species was discussed. The DEGs obtained by RNA-seq might be one of the explanations for the differences in leaf color among those three accessions.

## Figures and Tables

**Figure 1 plants-09-00748-f001:**
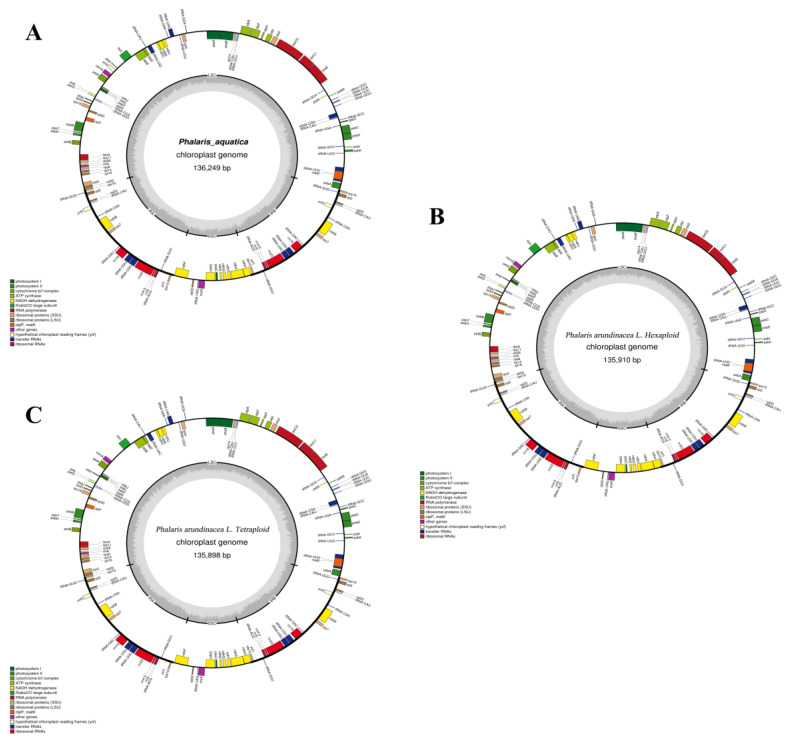
Gene maps of the *Phalaris* chloroplast genomes. Genes inside and outside of the circle are transcribed in clockwise and counterclockwise directions, respectively. The GC content and adenylate and thymine (AT) content are shown in dark gray and light gray, respectively. (**A**) *P. aquatica*; (**B**) hexaploid *P. arundinacea*; (**C**) tetraploid *P. arundinacea*.

**Figure 2 plants-09-00748-f002:**
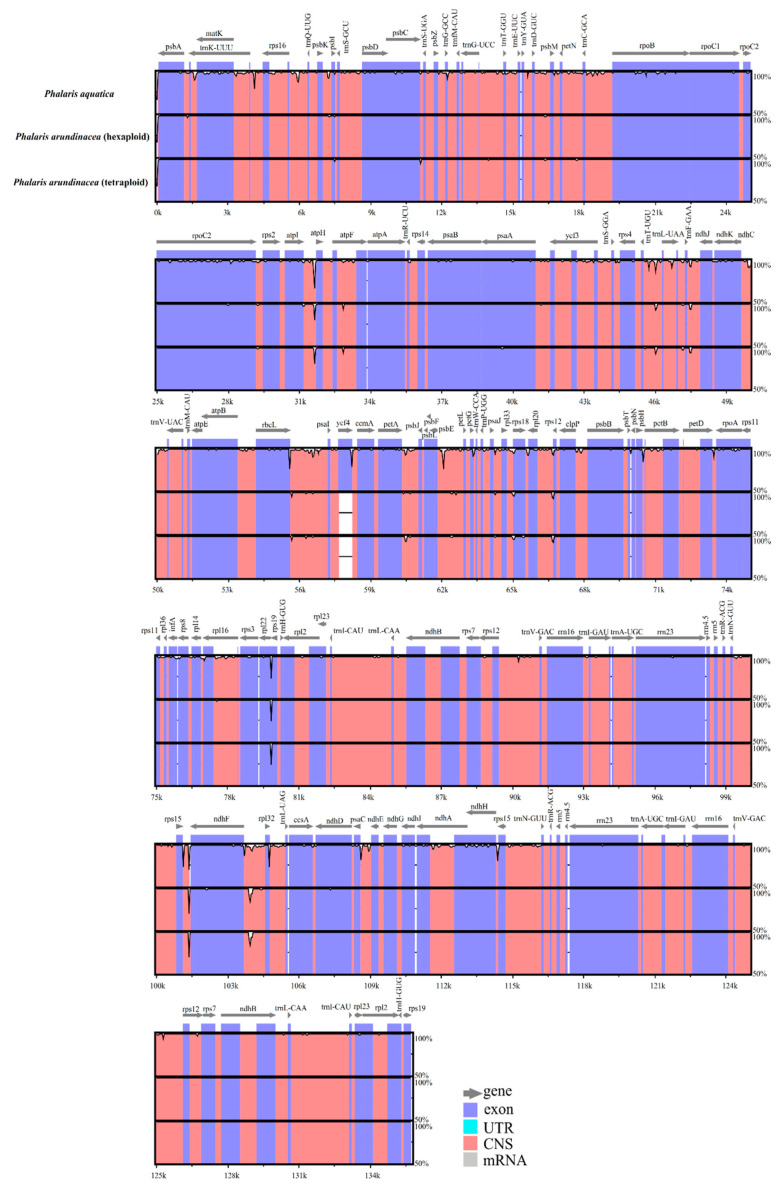
Alignment of the *Phalaris* chloroplast genome sequences. Exon, untranslated region (UTR), conserved noncoding sequences (CNS), and mRNA were color-marked. The *x*-axis and horizontal bars represent the coordinate and sequences similarity in the chloroplast genome, and the peaks indicate hotspot regions.

**Figure 3 plants-09-00748-f003:**
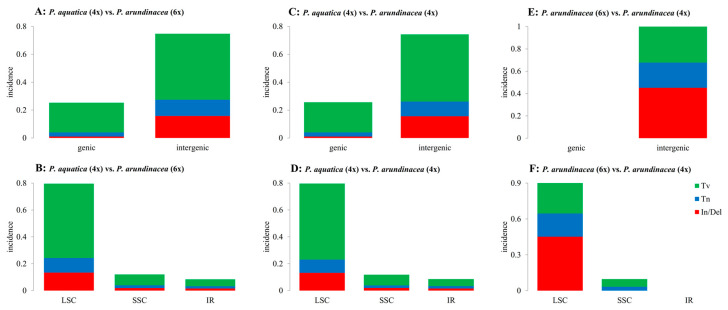
Summary of single nucleotide polymorphisms (SNPs) and indels. Tv, transversion; Tn, transition; In/Del, insertion and deletion; LSC, large single-copy region; SSC, small single-copy region; IR, inverted repeat region; (**A**,**B**), (**C**,**D**), and (**E**,**F**) reflect differences between *P. aquatica* (4x) vs. *P. arundinacea* (6x), *P. aquatica* (4x) vs. *P. arundinacea* (4x), and *P. arundinacea* (6x) vs. *P. arundinacea* (4x), respectively.

**Figure 4 plants-09-00748-f004:**
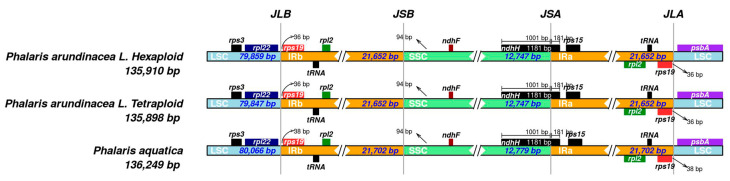
IRscope analysis of the three *Phalaris* cp genomes. JLB, junction of LSC and IRB region; JSB, junction of SSC and IRB region; JSA, junction of SSC and IRA region; JLA, junction of LSC and IRA region.

**Figure 5 plants-09-00748-f005:**
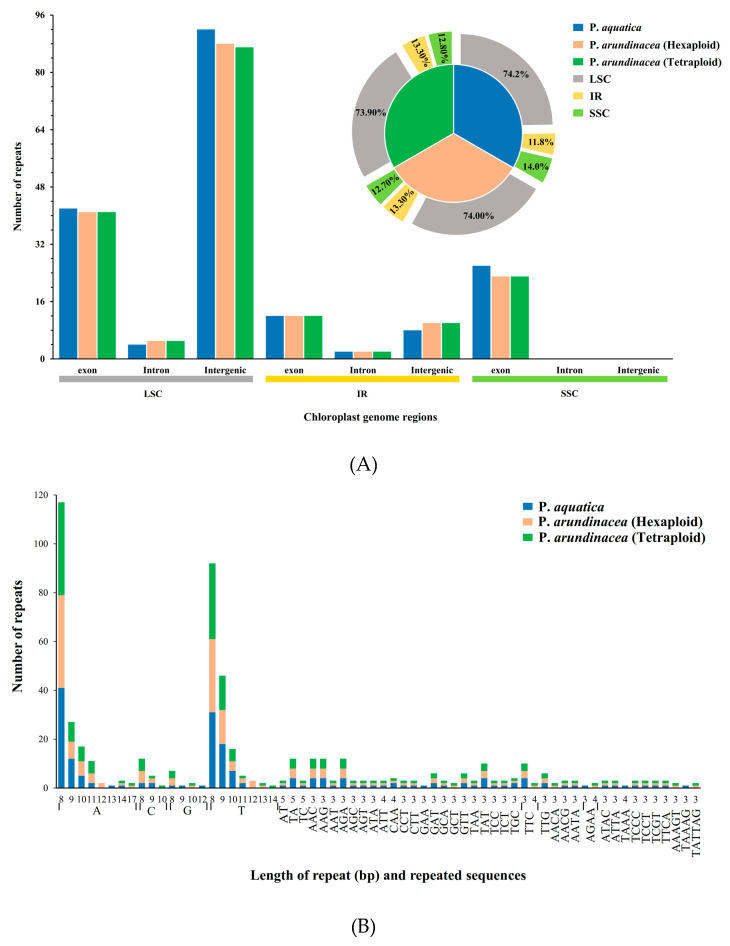
Simple sequence repeats (SSRs) and repeated sequences in the three *Phalaris* cp genomes. (**A**), SSRs in the different region of *Phalaris* cp genome; (**B**), motifs in the cp genome of *Phalaris*; (**C**), frequency of repeat types; (**D**), frequency of repeats length.

**Figure 6 plants-09-00748-f006:**
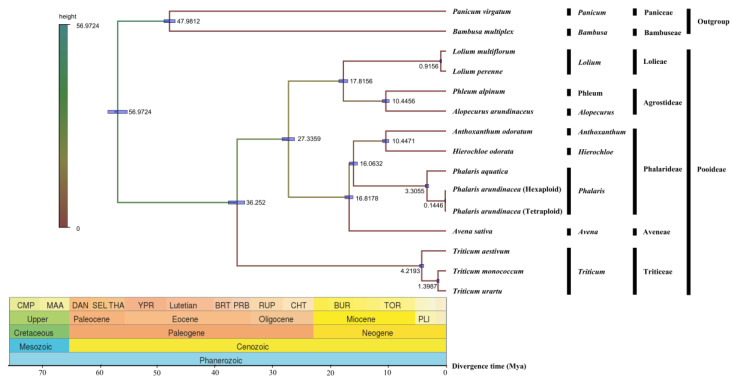
Divergence time among fifteen chloroplast genomes of seven tribes. Node values of the tree represent the average divergence time and blue bars of every node represent 95% credible interval.

**Figure 7 plants-09-00748-f007:**
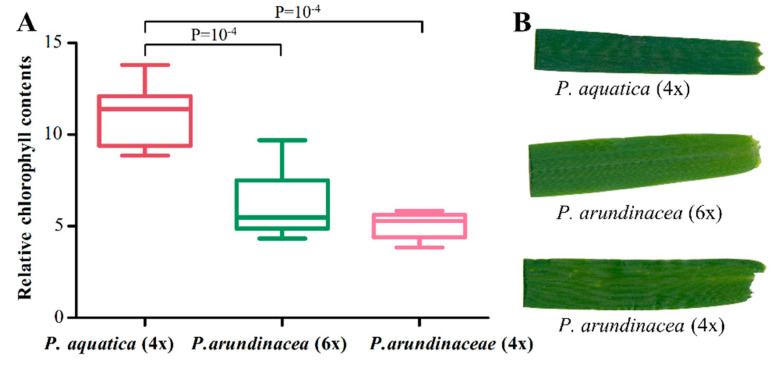
Comparison of the relative chlorophyll content of the leaves (**A**) and the leave color (**B**) of three *Phalaris* cultivars.

**Table 1 plants-09-00748-t001:** Comparison of the three *Phalaris* chloroplast (cp) genomes.

Species/Ploidies	Size (bp)				GC Content (%)				tRNA	rRNA	mRNA	Genes	Number of Genes Duplicated in IR
	cp	LSC	SSC	IR	cp	LSC	SSC	IR					
*P arundinacea* (4x)	135,898	79,846	12,746	21,653	38.52	36.45	32.88	44.00	28	4	72	104	19
*P arundinacea* (6x)	135,910	79,858	12,746	21,653	38.52	36.44	32.87	44.00	28	4	72	104	19
*P aquatica* (4x)	136,249	80,065	12,778	21,703	38.45	36.36	32.84	43.94	28	4	73	105	19

**Table 2 plants-09-00748-t002:** List of cp genes annotated in the three *Phalaris* cp genomes sequenced in this study.

Category	Function	Name of Gene					
Self-replication	Ribosomal RNA Genes	*rrn4.5*	*rrn5*	*rrn16*	*rrn23*		
	Transfer RNA genes	*trnA-ACG*	*trnL-CAA*	*trnV-GAC*	*trnH-GUG*	*trnA-GUU*	*trnA-UGC **
		*trnT-CGU **	*trnS-CGA **	*trnL-UAA **	*trnV-UAC **	*trnL-UUU **	*trnM-CAU #*
		*trnT-CCA*	*trnP-GAA*	*trnC-GCA*	*trnG-GCC*	*trnS-GCU*	*trnS-GGA*
		*trnT-GGU*	*trnT-GUA*	*trnA-GUC*	*trnL-UAG*	*trnA-UCU*	*trnS-UGA*
		*trnP-UGG*	*trnT-UGU*	*trnG-UUC*	*trnG-UUG*		
Ribosomal proteins (translation)	Small subunit of ribosome (SSU)	*rps2*	*rps3*	*rps4*	*rps7*	*rps8*	*rps11*
		*rps14*	*rps15*	*rps18*	*rps19*		
Transcription	Large subunit of ribosome (LSU)	*rpl2*	*rpl14*	*rpl16*	*rpl20*	*rpl22*	*rpl23*
		*rpl32*	*rpl33*	*rpl36*			
	RNA polymerase subunits	*rpoA*	*rpoB*	*rpoC1*	*rpoC2*		
	Translation initiation factor	*infA*					
Photosynthesis related genes	Large subunit of Rubisco	*rbcL*					
	Subunits of Photosystem I	*psaA*	*psaB*	*psaC*	*psaI*	*psaJ*	*ycf4* ^aq^
	Subunits of Photosystem II	*psbA*	*psbB*	*psbC*	*psbD*	*psbE*	*psbF*
		*psbH*	*psbI*	*psbJ*	*psbK*	*psbL*	*psbM*
		*psbT*	*psbZ*				
	Subunits of ATP synthase	*atpA*	*atpB*	*atpE*	*atpF **	*atpH*	*atpI*
	Cytochrome b/f complex	*petA*	*petB*	*petG*	*petL*	*petN*	
	C-type cytochrome synthesis gene	*ccsA*					
	Subunits of NADH dehydrogenase	*ndhA **	*ndhB **	*ndhC*	*ndhD*	*ndhE*	*ndhF*
		*ndhG*	*ndhH*	*ndhI*	*ndhJ*	*ndhK*	
Other genes	Maturase	*matK*					
	Protease	*clpP*					
	Chloroplast envelope membrane protein	*cemA*					
	Hypothetical protein	*ycf1*					
	Hypothetical open reading frames	*ycf2*					

Note: #, one gene copy in each IR; *, gene containing a single intron; genes in bold correspond to genes that are located in the IRs and hence are duplicated; ^aq^, genes that are particular for *P. aquatica.*

**Table 3 plants-09-00748-t003:** Intron-containing genes in the three *Phalaris* cp genomes.

Gene	*P. aquatica*				*P. arundinacea* (4x & 6x)			
	Location	Exon (bp)	Intron I (bp)	Exon II (bp)	Location	Exon I (bp)	Intron I (bp)	Exon II (bp)
*atpF*	LSC	160	818	407	LSC	160	826	407
*ndhA*	SSC	550	1020	539	SSC	550	1023	539
*ndhB*	IRA	775	712	758	IRA	775	712	758
*ndhB*	IRB	775	712	758	IRB	775	712	758
*trnS-CGA*	LSC	32	655	63	LSC	32	655	63
*trnT-CGU*	IRA	32	787	59	IRA	32	786	59
*trnT-CGU*	IRB	33	785	60	IRB	33	784	60
*trnL-UAA*	LSC	36	543	51	LSC	36	549	51
*trnV-UAC*	LSC	39	579	54	LSC	39	579	54
*trnA-UGC*	IRA	37	811	36	IRA	37	811	36
*trnA-UGC*	IRB	38	809	37	IRB	38	809	37
*trnK-UUU*	LSC	39	2465	37	LSC	39	2463	37

**Table 4 plants-09-00748-t004:** Accession information for three *Phalaris* cultivars.

Species	NPGS ID	Improvement Status	GenBank Accession
*Phalaris arundinacea* L. (tetraploid)	PI 272122	Cultivar	MT274594
*Phalaris arundinacea* L. (hexaploid)	PI 422031	Cultivar	MT274595
*Phalaris aquatica* L. (tetraploid)	PI 434985	Cultivar	MT274596

## Data Availability

The annotated chloroplast genomes of *Phalaris arundinacea* L. (tetraploid/4×), *P. arundinacea* L. (hexaploidy/6×) and *P. aquatica* L. (tetraploid/4×) have been deposited in the NCBI Genbank with the accession numbers MT274594, MT274595 and MT274596.

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
