# Peer review of "The Complete Chloroplast Genome Sequencing and Comparative Analysis of Reed Canary Grass (Phalaris arundinacea) and Hardinggrass (P. aquatica)"

_plants, 2020, doi:10.3390/plants9060748_

Round 1

Reviewer 1 Report

The manuscript “The complete chloroplast genome sequences of two Phalaris species reveal divergence among ploidies/species” is interesting, but from my point of view, it could be improved if included also wild representatives of both species.
There is a general incorrection when three species are referred (in several parts of the text). In fact, only two (P. aquatica and P. arundinacea) are studied. The two ploidies of the later species cannot be considered as two species, as they are cultivars.
Also, it should be clear in the text, not only in Material & Methods that only cultivars were used in the study.
The title, correctly, refers effectively to two species but using the expression ploidies/species does not seems correct. Please consider changing the title.
A brief explanation about the ploidy in these species would be welcome (e.g. is not clear whether P. arundinacea hexaploids occurs spontaneously in nature or are only man-made cultivars).

Author Response

All the modified parts have been marked in the manuscript with track changes. The responses to the reviewer's questions are as follows.

Question 1: The manuscript “The complete chloroplast genome sequences of two Phalaris species reveal divergence among ploidies/species” is interesting, but from my point of view, it could be improved if included also wild representatives of both species.

Answer: Thanks for the reviewer's suggestion. The three sequenced Phalaris accessions were all NPGS collections. Indeed, the manuscript could be improved if wild representatives were included. Given that the character of cp genome is maternal inheritance, there was little variation among wild germplasm and cultivars, even among different genus of the Poaceae Family. Therefore, we think it’s not profitable to sequence wild accessions of the two Phalaris species. In addition, most of the Phalaris cultivars were selected from wild germplasms by artificial domestication. Of course, it’s necessary to study the phylogeny of wild Phalaris accessions, cp SSRs developed from cp genomes are powerful tool to reveal the genetic diversity and variation, and that's what we're going to do next.

Question 2: There is a general incorrection when three species are referred (in several parts of the text). In fact, only two (P. aquatica and P. arundinacea) are studied. The two ploidies of the later species cannot be considered as two species, as they are cultivars. Also, it should be clear in the text, not only in Material & Methods that only cultivars were used in the study.

Answer: Thanks for the reviewer's suggestion. Indeed, the statement ‘three species’ was less rigorous, so we have amended it in the whole manuscript. We have also made this clear in the text (Table 4, line 368) that only cultivars were used in the study.

Question 3: A brief explanation about the ploidy in these species would be welcome (e.g. is not clear whether P. arundinacea hexaploids occurs spontaneously in nature or are only man-made cultivars).

Answer: Thanks for the reviewer's suggestion. As mentioned in the introduction section (line 77), the Phalaris genus presents a potentially valuable case study for speciation in Poaceae due to its key role of polyploidy in its evolution, so the formation of hexaploids and tetraploid P. arundinacea was spontaneously in nature while not only man-made cultivars.

Reviewer 2 Report

Title is very wrongly stated and misleading – “reveal divergence among 4 ploidies/species” – dose not make any sense.

Abstract

“The 22 species in Phalaris L. genera……” the sentence need to be rewrite

(tetraploid/4x and hexaploid/6x) – 4x and 6x need to remove, only tetraploid and hexaploid is sufficient.

 Authors need to rewrite the entire Abstract to make it clearer about the aim of study, obtained results and overall message.

Introduction

Please make sure about the “quadripartite structure”

Line 91 – need to correct the flow of information, authors suddenly changed the topic from cp genome to RNAseq. It should be start with new para, and that too after concluding the preceding para.

In title, cp genome sequencing of  only two mentioned but in the abstract and introduction authors have mentioned sequencing of three species. This is true that the three genomes are belongs to two species but it need to make clear either in title or in the abstract.

Results

Authors need to make sentence simpler as well as reduce the descriptive writing. For instance -  “The pair of inverted repeats (IRA and IRB) regions of the three species are 106 21,653 bp, 21,653 bp and 21,703 bp, separated by a LSC region (79,846 bp, 79,858 bp and 80,065 bp) 107 and a SSC region (12,746 bp, 12,746 bp and 12,778 bp) (Table 1 and Fig 1)”

“The 72 common mRNA genes consist of 24 unique transcription- and translation-related genes” – this is confusing, what are commons and unique.

Figure 5 and 7 can be moved to supplementary.

I am not sure how relevant is the RNAseq data here in this MS. I suggest to remove and may make separate article for RNAseq study.

Discussion

Author’s have repeated results in the discussion section. Sometime it is necessary to do so, but here authors need to remove most of such result and focus on points authors want to discuss.

Author Response

All the modified parts have been marked in the manuscript with track changes. The responses to the reviewer's questions are as follows.

Question 1: Title is very wrongly stated and misleading – “reveal divergence among 4 ploidies/species” – dose not make any sense.

Answer: Thanks for the reviewer's question. We have changed the title into “The complete chloroplast genome sequencing and comparative analysis of Phalaris arundinacea and P. aquatica.”

Question 2: “The 22 species in Phalaris L. genera……” the sentence needs to be rewrite.

(tetraploid/4x and hexaploid/6x) – 4x and 6x need to remove, only tetraploid and hexaploid is sufficient.

Answer: Thanks for the reviewer's question. We have rewritten the sentence “The 22 species in Phalaris L. genera……” into “There were 22 species in the Phalaris genera that distribute almost all over the temperate regions of the world” in line 24-25. In addition, “4x and 6x” in the description “tetraploid/4x and hexaploid/6x” was removed in line 26 and Table 4.

Question 3: Authors need to rewrite the entire Abstract to make it clearer about the aim of study, obtained results and overall message.

Answer: Thanks for the reviewer's question. Our aim in this study was to reveal the chloroplast genome differences between the P. arundinacea and P. aquatica species and divergence between tetraploid and hexaploid P. arundinacea, and we also amended the abstract as much as possible, the revisions was highlighted using the track changes in the abstract.

Question 4: Please make sure about the “quadripartite structure”

Line 91 – need to correct the flow of information, authors suddenly changed the topic from cp genome to RNAseq. It should be start with new para, and that too after concluding the preceding para.

Answer: Thanks for the reviewer's question. The terminology “quadripartite structure” was usually used to describe the classical chloroplast genomes structures, that was also mentioned in many other articles, such as “Chloroplast genomes: diversity, evolution, and applications in genetic engineering” in Genome Biology (2016), “A high level of chloroplast genome sequence variability in the Sawtooth Oak Quercus acutissima” in International Journal of Biological Macromolecules (2020), “Complete Chloroplast Genome Sequence and Phylogenetic Analysis of Quercus acutissima” in International Journal of Molecular Sciences (2018) and so on. Indeed, it’s sudden to change the topic from cp genome to RNAseq, we have started with new para to draw forth the RNAseq in line 94.

Ref:

[1] Daniell, H.; Lin, C.S.; Yu, M.; Chang, W.J. Chloroplast genomes: diversity, evolution, and applications in genetic engineering. Genome Biol. 2016, 17(1), 134.

[2] Zhang, R.S.; Yang, J.; Hu, H.L; Xia, R.X.; Qin, L. A high level of chloroplast genome sequence variability in the sawtooth oak quercus acutissima. International Journal of Biological Macromolecules. 2020, 152, 340-348.

[3] Li, X.; Li, Y.F.; Zang, M.Y.; Li, M.Z.; Fang, Y.M. Complete Chloroplast Genome Sequence and Phylogenetic Analysis of Quercus acutissima. International Journal of Molecular Sciences. 2018, 19, 2443.

Question 5: In title, cp genome sequencing of only two mentioned but in the abstract and introduction authors have mentioned sequencing of three species. This is true that the three genomes are belongs to two species but it need to make clear either in title or in the abstract.

Answer: Thanks for the reviewer's question. In this study, we sequenced three cp genomes of Phalaris species, that is P. aquatica L., tetraploid P. arundinacea L. and hexaploid P. arundinacea L., while the tetraploid P. arundinacea L. and hexaploid P. arundinacea L. are both belonged to P. arundinacea L. Actually, only two species (P. arundinacea L. and P. aquatica L.) were sequenced in this study. So, we have checked and modified relevant content in the whole manuscript to make it clear.

Question 6: Authors need to make sentence simpler as well as reduce the descriptive writing. For instance - “The pair of inverted repeats (IRA and IRB) regions of the three species are 106 21,653 bp, 21,653 bp and 21,703 bp, separated by a LSC region (79,846 bp, 79,858 bp and 80,065 bp) 107 and a SSC region (12,746 bp, 12,746 bp and 12,778 bp) (Table 1 and Fig 1)”

Answer: Thanks for the reviewer's suggestion. We have changed sentence “The pair of inverted repeats (IRA and IRB) regions of the three species are 106 21,653 bp, 21,653 bp and 21,703 bp, separated by a LSC region (79,846 bp, 79,858 bp and 80,065 bp) 107 and a SSC region (12,746 bp, 12,746 bp and 12,778 bp) (Table 1 and Fig 1)” into “The pair of inverted repeats (IRA and IRB) regions of the three cp genomes are 21,653 bp, 21,653 bp and 21,703 bp, separated by a LSC region  and a SSC region (Table 1 and Fig 1)” in line 112-114. We have also reduced the descriptive writing as much as possible in the whole manuscript and highlighted using the track changes, such as line 123-128, line 207, line 208 and line 278-280.

Question 7: “The 72 common mRNA genes consist of 24 unique transcription- and translation-related genes” – this is confusing, what are commons and unique. Figure 5 and 7 can be moved to supplementary.

Answer: Thanks for the reviewer's question. The common genes refer to genes that were existed in all the three Phalaris cp genomes. Genes in the IRs region are duplicated, so when count the number of genes, we only count once for each gene, that is a unique gene. And the use of “common” and “unique” was also existed in many others articles such as “Chloroplast genomes: diversity, evolution, and applications in genetic engineering” in Genome Biology (2016) and “Comparative Analysis of the Complete Chloroplast Genome Sequences of Three Closely Related East-Asian Wild Roses (Rosa sect. Synstylae; Rosaceae)” in genes (2019). Furthermore, Figure 5 and 7 have been moved to supplementary.

Ref:

[1] Daniell, H.; Lin, C.S.; Yu, M.; Chang, W.J. Chloroplast genomes: diversity, evolution, and applications in genetic engineering. Genome Biol. 2016, 17(1), 134.

[2] Jeon, J.H.; Kim, S.C. (2019). Comparative analysis of the complete chloroplast genome sequences of three closely related east-asian wild roses (rosa sect. synstylae; rosaceae). Genes. 2019, 10(1), 23.

Question 8: I am not sure how relevant is the RNAseq data here in this MS. I suggest to remove and may make separate article for RNA-seq study.

Answer: Thanks for the reviewer's suggestion. The result indicated only minor difference among the three studied cp genomes. In order to further illuminate the accurate relationships or differences among the three genomes, RNA-seq was adopted to look into the chloroplast gene expressions by mapped the RNA-seq reads of each accession to their own cp genome. As mentioned in the introduction, the fully transcribed character of cp genome was proved in previous studies, that further proved the feasibility and necessity of RNAseq. As expected, significant gene expression difference was detected in several cp genes. What’s more, the correlation between genotype and phenotypes was also a research hotspot, that was preliminary confirmed by the leaf color-related genes in the cp genomes. Of course, we will continue to focus on the study of chloroplast gene function of Phalaris.

Question 9: Authors have repeated results in the discussion section. Sometime it is necessary to do so, but here authors need to remove most of such result and focus on points authors want to discuss.

Answer: Thanks for the reviewer's question. We have simplified the discussion and highlighted using the track changes in line 285-288, line 292-293, line 328-331, line 350-351.

Round 2

Reviewer 2 Report

The revised version looks significantly improved. Few minor issues are there which need to be addressed. For instance, most of the subheadings are not improved. for example -"Cp genome characteristics of three Phalaris cp genomes" - here CP is redundant and also better not to use abbreviations in the subheadings.

Author Response

All the modified parts have been marked in the manuscript with track change. The responses to the reviewer's questions are as follows.

Question1: The revised version looks significantly improved. Few minor issues are there which need to be addressed. For instance, most of the subheadings are not improved. for example -"Cp genome characteristics of three Phalaris cp genomes" - here CP is redundant and also better not to use abbreviations in the subheadings.

Answer: Thanks for the reviewer’s suggestion. We have checked and improved all the subheadings and changed “2.1 Genomic features of three Phalaris cp genomes” into “2.1 Genomic features of three Phalaris chloroplast genomes” in line 109, “2.2. Variation among three cp genomes” into “2.2. Variation among three chloroplast genomes” in line 144, “2.8 Diversity and expression of cp genes” into “2.8 Diversity and expression of chloroplast genes” in line 258, “3.1 Cp genome characteristics of three Phalaris cp genomes” into “3.1 Chloroplast genome characteristics of three Phalaris cultivars” in line 283, “3.5 Gene expression in cp genomes” into “3.5 Gene expression in chloroplast genomes” in line 349, “4.2 Cp genome assembly and annotation” into “4.2 Chloroplast genome assembly and annotation” in line 376, “4.3 Multiple cp genome alignments” into “4.3 Multiple chloroplast genome alignments” in line 389, “4.6 Analysis of Ka/Ks” into “4.6 Analysis of non-synonymous/synonymous substitution” in line 414.